# Development of Smart Material Identification Equipment for Sustainable Recycling in Future Smart Cities

**DOI:** 10.3390/polym17040462

**Published:** 2025-02-10

**Authors:** Gaku Manago, Tadao Tanabe, Kazuaki Okubo, Tetsuo Sasaki, Jeongsoo Yu

**Affiliations:** 1Graduate School of International Cultural Studies, Tohoku University, Sendai 9808576, Japan; kazuaki.okubo.d5@tohoku.ac.jp (K.O.); jeongsoo.yu.d7@tohoku.ac.jp (J.Y.); 2Department of Engineering and Design, Shibaura Institute of Technology, Tokyo 1358548, Japan; 3Graduate School of Medical Photonics, Shizuoka University, Hamamatsu 4328011, Japan; sasaki.tetsuo@shizuoka.ac.jp

**Keywords:** smart equipment, sustainable recycled polymers, waste plastic, sustainable circular economy, technology and innovation

## Abstract

Waste recycling is critical for the development of smart cities. Local authorities are responsible for the disposal of waste plastics, but the extent of material recycling is insufficient, and much of the waste generated is incinerated. This conflicts with the trend of decarbonisation. Of particular note are the effects of the COVID-19 pandemic, during and after which large quantities of waste plastics, such as plastic containers and packaging, were generated. In order to develop a sustainable smart city, we need an effective scheme where we can separate materials before they are taken to the local authorities and recyclers. In other words, if material identification can be performed at the place of disposal, the burden on recyclers can be reduced, and a smart city can be created. In this study, we developed and demonstrated smart material identification equipment for waste plastic materials made of PET, PS, PP, and PE using GaP THz and sub-THz wavelengths. As basic information, we used a GaP terahertz spectrometer to sweep frequencies from 0.5 THz to 7 THz and measure the spectrum, and the transmittance rate was measured using the sub-THz device. The sub-THz device used a specific frequency below 0.14 THz. This is a smaller, more carriable, and less expensive semiconductor electronic device than the GaP. Moreover, the sub-terahertz device used in the development of this equipment is compact, harmless to the human body, and can be used in public environments. As a result, smart equipment was developed and tested in places such as supermarkets, office entrances, and canteens. The identification of materials can facilitate material recycling. In this study, we found that measuring devices designed to identify the PET and PS components of transparent containers and packaging plastics, and the PP and PE components of PET bottle caps, could effectively identify molecular weights, demonstrating new possibilities for waste management and recycling systems in smart cities. With the ability to collect and analyse data, these devices can be powerful tools for pre-sorting.

## 1. Introduction

Waste plastic emissions have increased since the advent of the COVID-19 pandemic, precipitating a need for the further recycling of waste plastics as a measure to reduce carbon dioxide emissions [1]. In smart cities, AI and the IoT are being introduced into the recycling sector [2]. For example, IoT devices are being used to optimise waste collection routes to reduce fuel consumption and costs. Demonstrations have also been made in 18 locations in Tokyo; one example involved the installation of small-household-appliance collection boxes with built-in weight sensors and other devices, which allow Japanese companies to remotely check the statuses of boxes [3]. This system is known as an urban mine, and the medals awarded at the Tokyo Olympics and Paralympics were made from these resources, as introduced by the media. These digital and physical devices are data-managed and useful for smart cities [4,5,6]. In this way, the presence of accessible recycling facilities in towns can promote smart cities. In Japan, some supermarkets and convenience stores collect waste plastic through small resource collection units [7]. PET bottles in particular are given priority (via a loyalty program) based on their abundance. In this case, the caps and labels are removed, and the bottles are put into the units one by one. Contributions to this loyalty program can be redeemed in the form of a discount coupon when shopping at the facility. Establishments that do not have such resource collection units have simple boxes with labels attached to them. Transparent containers, PET bottles, and foamed plastic are collected, and customers can drop off waste plastic, which can be visually sorted, in person. These actions constitute supermarkets and convenience stores’ corporate efforts to be environmentally friendly, but they are also a means of attracting customers. The customers clean and wash plastic waste stored at home and dispose of it in these boxes. In Japan, depending on the municipality, denizens must buy designated refuse bags for the disposal of plastic waste. Thus, if we can dispose of waste in a resource collection box at a supermarket, we can reduce the cost of waste bags. In other words, the installation of small plastic resource collection units is a win-win situation for the businesses and the consumers who install them. The study conducted by Jaid et al. also suggested that the IoTisation of rubbish bins promotes the development of smart cities [8,9]. The waste sector in smart cities can be described in terms of a smart environment [10]. Therefore, the installation of small resource collection units will promote recycling by the people living in a town, thus creating a smart, environmental town. In fact, most municipalities must deal with the heavy burden of separating recyclable materials because there are so many types of materials mixed in with household waste. On the other hand, several manufacturers and large supermarkets want to collect clean and single-type plastic waste directly from residents. Through big data analysis, direct collection routes can effectively improve recycling efficiency and allow the accurate analysis of consumer behaviour. Based on concrete evidence, a new recycling system and network can help reduce the inefficiency and excessive cost of recycling in each municipality. In addition, it can also be utilised to develop effective material identification schemes, making it a useful means of developing a sustainable, smart city.

Therefore, sorting waste plastics by material at the point of disposal could reduce the burden on the municipalities and recyclers who conduct the sorting and make for a smarter city. Therefore, we have developed material identification equipment for transparent plastic containers, packaging (PET and PS), and plastic bottle caps (PP and PE), which are collected and identified in supermarkets to promote the development of smart cities. We designed smart material identification equipment that can be installed in supermarkets and small spaces, constructed it (constituting an original system), and confirmed its effectiveness in laboratory experiments using actual samples. In addition, a demonstration experiment was also conducted to evaluate the system (Figure 1).

The originality of this research lies in the fact that while near-infrared instruments (hereafter termed NIR) have been used in previous plastic identification devices that can be installed in confined spaces such as supermarkets, we used sub-THz instruments. Furthermore, it is important to note that the characteristics of the material are determined and identified based on spectral data ranging from the sub-THz spectrum to that of gallium phosphide (hereafter referred to as GaP THz). Another feature of sub-terahertz instruments is that they allow identification regardless of the colour of the plastic, making them useful for identifying plastic containers and bottle caps made of PET, PS, PP, and PE. It is hoped that the increasing popularity of this technology will lead to a more sustainable and efficient society.

## 2. Research Question and Previous Studies

### 2.1. The Wastage of Containers and Packaging

The quantities of waste plastic containers and packaging have increased since the advent of the COVID-19 pandemic [11] due to the increase in hygiene awareness and the need to prevent infection. In addition, plastic containers and packaging can be sealed and disposable, thus increasing user safety and helping to prevent infection. Furthermore, the quantities of plastic containers and wraps have increased to ensure that food is taken home safely, such as in the case of takeaway and deliveries, due to the impact of the coronavirus [12]. The transparent containers of disposable lunchboxes, especially for those whose contents should be visible, are usually made of PET and PS [13]. The compositional material of these transparent containers cannot be determined visually. The recycling boxes that are resource-collected in supermarkets in Japan are simply labelled as transparent container collection boxes, and they are not specifically separated by material. Therefore, in order to develop a smart city, it is desirable to separate each material in supermarkets.

### 2.2. Bottle Cap Waste

Bottled beverages had a significant impact on public health awareness and consumption behaviour during the COVID-19 pandemic, during which citizens generally purchased and used individually sealed beverage containers, such as plastic bottles, in order to avoid the use of shared products [14]. Shared water servers and public beverages were no longer available, and individuals prepared their own drinks. Therefore, plastic bottles were used, and beverages were bought in bulk due to their long shelf lives. There was also a growing demand for disaster reserves, which people kept in their homes at all times. The body of a PET bottle is made of PET, while the cap is composed of polyethylene or polypropylene [15]. Machines designed to collect PET bottles are installed in many smart cities [16]. In Europe, recommendations suggest that PET bottles and caps should be collected at the same time, while in Japan, recommendations suggest that PET bottles and caps should be collected separately [17]. As PET bottles can be traded at a high price, shops are awarded for putting empty PET bottles into their collection machines. This deposition system indicates a good relationship between a smart city and a recycling system. In Japan, bottles and caps are separated, but loyalty is not given to caps. Therefore, collecting waste caps made of pure plastic of a single type and awarding points for each donation would change citizens’ recycling awareness. Previous studies have shown that plastic bottle caps can be identified and sorted into PP and PE, but no smart equipment has yet been developed to identify caps in small spaces.

### 2.3. Research on Technology and Waste

NIR instruments are used in plastic material measurements. Near-infrared light is widely used in the spectroscopy of plastics. It is inexpensive and, compared to X-rays, less harmful to human health [18]. However, the reflectance is low because plastic, whether transparent or not in the visible region, has a low refractive index in the NIR region. Moreover, the inability of NIR instruments to identify black plastics is a problem [19]. Thus, containers and plastic bottle caps that are black cannot be detected. Therefore, we have carried out plastic identification studies using terahertz radiation [15]. Moreover, this technology is easy to use, so it will be helpful in solving the waste problem in developing countries via international cooperation [20].

Terahertz spectroscopy is a technique that can provide information on the dielectric properties of materials at terahertz frequencies and be used for identification [21]. This technique can be used to accurately identify differences in molecular weight, an ability that is expected to improve the accuracy of material recycling and increase the re-use rate of waste plastics. The use of the recently developed GaP terahertz light source (0.5–7.5 THz) enables a deeper understanding of materials, additives, and degradation conditions in terahertz-wave technology spectroscopy applications [22]. Terahertz laser spectrometers have not only been used to detect defects in pharmaceuticals, but also to research and develop applications for the material sorting of plastics. However, as it is difficult to sort materials based on the transmittance and reflectance of a single monochromatic light source; a polychromator system, which combines multiple light sources, is used to realise high speed and high sorting accuracy. In this study, we used a GaP terahertz light source in the terahertz band as a monochromatic light source and semiconductor device transmitters, such as Tunnel Injection Transit Time (TUNNETT) diodes, GUNN diodes, and Impact Avalanche and Transit Time (IMPATT) diodes, in the sub-terahertz band of fixed-frequency light sources for further frequency extension. A semiconductor device transmitter can be combined with a Schottky diode detector to build a system using only semiconductor devices. In addition, semiconductor devices are smaller, simpler, more stable, and more durable than terahertz-wave devices using GaP crystals as a light source. Therefore, plastic materials can be sorted in the THz band, but as GaP systems are large and expensive, it is necessary to develop a sorting system using semiconductor devices in the sub-THz band, which will need to be smaller and cheaper. However, the sub-terahertz band has strong radio wave characteristics, so when applied to spectroscopy systems, the effects of interference are significant, requiring measurement techniques such as standing wave methods. In addition, the sub-terahertz-to-terahertz band corresponds to the frequency of intermolecular vibrations between molecular chains; therefore, in material identification, the orientation of the molecular chains affects the measurement [23].

### 2.4. Research on Smart Cities and Waste

As mentioned above, research has been carried out on waste recycling and identification technology. In this field, identification technology is an area of engineering research, while informatics disciplines are needed to determine the materials. The field of waste has so far made use of IoT, AI, and opportunity learning [24]. The IoT sector has been an especially popular subject in this regard. In the waste management sector, there are IoT studies involving infrastructure in urban environments [25] and resource recycling [26]. In addition, the IoT can maximise a variety of services in the waste, infrastructure and transport, welfare and healthcare [27], sanitation and water management [28], and education sectors [29]. The device we have developed also collects data in advance and matches them with a database in the cloud to predict material qualities.

In this study, a system for sorting plastic materials based on spectral information obtained from GaP terahertz light sources and sub-terahertz electronic devices was constructed, as shown in Figure 2. This system is based on semiconductor devices, which are compact and relatively inexpensive, making them suitable for future smart cities. These devices need to be constructed to avoid electromagnetic interference, and the measurement system needs to be unaffected by anisotropy due to the orientation of molecular chains. Based on spectroscopic data from conventional GaP terahertz light sources, we designed, constructed, and realised a measurement system in the sub-terahertz wave band. The usefulness of this device was confirmed through tests using real plastic samples of disposable container packaging (PET/PS) and plastic bottle caps (PP/PE). Thus, instead of using NIR, which is traditionally used in plastic identification, we not only designed a sorting system using sub-THz and THz radiation, but also constructed it, serving as a unique system, and confirmed its usefulness through measurements based on practical samples.

## 3. Experimental

### 3.1. GaP Terahertz Spectroscopy Measurement System

A GaP THz source was used to obtain basic data that provided scientific evidence. The terahertz spectroscopy measurement system, using a GaP light source, can sweep from 0.5 THz to 7 THz, and it uses a laser as the excitation light source, which is mainly adapted for spectral measurements, so it is desirable to perform measurements indoors in a laboratory or some other environment that is not affected by temperature and other environmental conditions. The operating principle of this device is based on terahertz wave generation via differential frequency mixing, and the purity of the generated terahertz wave is due to the addition of the frequency purity of the pumping infrared laser. The GaP light source can generate pulses, even nanosecond pulses, which are not constricted by the Fourier limit, and operate in CW mode, allowing measurements with high resolution. Furthermore, transmission and reflection measurements are possible. A pyroelectric Deuterium Tri-Glycine Sulphate (DTGS) detector element operating at room temperature was used as the detector.

### 3.2. Design and Construction of the Sub-Terahertz-Band Measurement System 

GUNN diodes, IMPATT diodes, and resonant-tunnelling diodes were used as light sources in the sub-terahertz band. The frequency bands that can be generated are 0.075 THz, 0.1 THz, and 0.14 THz. These can be transmitted in CW mode by applying a specific voltage. The sample was irradiated with these bands, and the transmitted intensity was determined. Schottky barrier diodes were applied to the detector, and horn antennas of the appropriate size for the respective band were used for the source and the detector to increase the waveguide efficiency in free space and in the device. The measurement system has a transmission arrangement for sample detection. The physical properties of the material, such as thickness, may affect the measurements due to the interference properties of terahertz waves. Therefore, the beam size and incidence conditions were appropriately adjusted.

## 4. Measurement Samples

The container and packaging plastics were divided into PET and PS, and PET bottles were divided into PET, PP, and PE.

### 4.1. PET/PS Material

The PET/PS measurement sample was a disposable transparent container from a household. After collecting the disposable containers, we rinsed their surfaces with water and used the containers as samples. These single-material samples were all transparent containers, whose size and thickness varied per sample. The samples were larger than the terahertz and sub-terahertz measurement beam sizes, which ranged from 10 cm to 15 cm square and 0.1 to 0.2 mm thick.

### 4.2. PP/PE Material

The samples measured for the PP/PE were PET bottle caps recycled in Japan. Samples of PET bottle caps with an outer diameter of 28 mm and heights of 7.5, 8, 13.5, and 14.5 mm were used to obtain basic data on the samples without inner lids. All the samples were white.

## 5. Results of THz Spectroscopy Experiments Using a GaP Light Source

### 5.1. Molecular-Weight-Level Identification of PET and PS

Figure 3 shows the THz spectra of the PET and PS disposable containers at room temperature obtained using a GaP crystal as the light source. As seen in Figure 3a, PET has a large absorption peak around 4.1 THz [30]. As shown in Figure 3b, PS has an absorption peak of around 2.5 THz. An absorption peak at 4.1 THz was observed for both film thicknesses; it is known to correspond to the bonding strength of the whole crystal. Based on its intensity, the melting point can be estimated in a non-contact manner. Such spectral properties in the high-frequency terahertz band also have an effect in the sub-terahertz band, where the frequencies are close, and electronic devices operating in the sub-terahertz band can be used to make compact and portable systems.

### 5.2. Molecular-Weight-Level Identification of PP and PE

Figure 4 shows the terahertz spectra of PP and PE at room temperature obtained using a GaP crystal as a light source. Figure 4a,b show the measured terahertz spectra of a PP plate and a PE plate as standard spectra, and Figure 4c depicts the terahertz spectrum of a plastic bottle cap. As shown in Figure 4, PP has absorption peaks around 3 THz and 5 THz. PE has an absorption peak of around 2.1 THz. For the plastic bottle caps, absorption peaks were observed at around 3 THz for PP and around 2.1 THz for PE. Furthermore, the difference in transmittance between PP and PE in the sub-terahertz band below 0.5 THz is large, indicating that the sub-terahertz region can be easily distinguished. The spectral interference in the background is dependent on film thickness and frequency and appears under conditions where the absorption coefficient in that frequency band is small. Incidentally, the period increases with a decrease in film thickness.

## 6. Results of the Design and Construction of an Identification System Using Sub-THz Devices

### 6.1. PET/PS Identification and the Proposed Sorting System

For transparent PET and PS plastic containers, two types of terahertz waves were set from above the sample stage, and the transmittance was calculated by measuring the transmittance intensity. The transparent container samples were placed with their tops open. Figure 5 shows the transmittances of the disposable container packaging made of PET and PS materials as measurement targets, reflecting their dielectric properties. Six and eight different samples of PET and PS material were used, respectively. We measured the transmittances at frequencies of 0.075 and 0.1 THz. The frequency dispersion can be seen in the two-dimensional plots with the transmittance of 0.075 THz and 0.1 THz as the parameters. The results showed that PET had low transmittance at 0.075 THz; PS had high transmittance at 0.075 THz, with a variation at 0.1 THz; and PS had low transmittance at 0.075 THz and 0.1 THz, with a variation at 0.1 THz. These variations were considered to be due to interference in the measurement as well as scattering caused by irregularities on the surfaces of the containers in the measurement area. The 0.075 THz frequency has a wavelength of 4 mm, which is less varied than the shorter wavelengths of 0.1 THz. Therefore, the two sub-THz frequencies are suitable for identifying PS and PET containers.

In addition, as shown in Figure 6, z-scores were used for the evaluation of this equipment. The z-scores were calculated for each sample based on the following formula:(1)$$z_{ij}=\frac{x_{ij}-{\bar{x}}_{j}}{\sigma_{j}}$$

Here, ${\bar{x}}_{j}$ represents the mean transmittance at frequency j, $\sigma_{j}$ is the standard deviation, and $x_{ij}$ is the transmittance of sample i at frequency j.

It is evident that for the PET samples, the z-scores calculated from transmittance at frequencies of 0.1 THz and 0.075 THz are consistently negative, whereas the z-scores tend to be positive for the PS samples. Based on the observation above, it can be said that when trying to determine whether a sample is made of PET or PS, samples with transmittance values measured at 0.1 THz or 0.075 THz that are lower than the overall average can be classified as PET, while those with higher transmittance values can be classified as PS. The difference can be distinguished with two THz devices. These data can be stored in the cloud and be used as collated data for identification.

### 6.2. Identification of PE/PP: Design and Construction of Identification Equipment Using Sub-THz Devices

The caps were removed from the bottles. A sub-THz device was placed above the cap level, and the transmittance was measured. The sub-THz device operated at a frequency of 0.14 THz. The caps were placed on the stage with the top open for measurement. Figure 7 and Table 1 show the transmittance values proportional to the intensity of the transmitted sub-terahertz wave when the tops of the PP and PE caps were irradiated with 0.14 THz sub-terahertz waves: 46.9–62.7% for PP and 67.9–77.6% for PE. Therefore, based on the significant difference in the transmittance values, a frequency of 0.14 THz is suitable for identifying PE and PP caps.

## 7. Results: Development and Sorting Test

### 7.1. Development and Demonstration of Smart Identification Equipment for PET and PS Containers

As an applicational experiment, a sub-THz device capable of making measurements was constructed for use outside. Sub-terahertz band devices are compact and harmless to humans, so the constructed device could be carried from the laboratory to a public environment for measurement. The experiments showed that the two sub-THz devices can effectively identify PET and PS containers. Therefore, we developed a piece of smart equipment that can be installed in small spaces in a supermarket or convenience store for demonstrating experiments using the same frequency. As shown in Figure 8a, two sets of sub-THz oscillator sources and detectors were arranged at a 45-degree angle in a 750 mm × 500 mm × 1000 mm box. Their light was focused at the centre of the box. The oscillators were mounted at the 0.075 THz and 0.1 THz positions starting from the top left. The transparent container to be measured was set at the sub-terahertz focal point, and the transmission intensity was measured. The intensity was converted using an AD converter and sent to a PC. This PC contained a transmittance intensity database that we had already created. As a result of matching, the system displays predicted materials with approximate properties. Based on the display results, the user can sort the materials by hand. In the future, we would like to automate it using a conveyor or similar device. The design of the measurement machine for the PET/PS container was based on a schematic.

This smart identification device that can distinguish between PET and PS using sub-THz waves was installed in a supermarket in Japan. Figure 8b shows the demonstration. The target samples were disposable containers whose surfaces had been cleaned at home by customers. The disposable containers were fed into the device by the customers themselves, and the device immediately displayed the results regarding material quality. A multinomial logistic regression model was applied to the material identification of used plastics, and the result was a 93% identification accuracy. The difference in the results between the experiments may be due to the contamination and shapes of the containers. This design also allows customers to easily feed samples of waste plastic into the machine (Figure 8b).

### 7.2. Development and Demonstration of Smart Identification Equipment for PP and PE Bottle Caps 

As an applicational experiment, a sub-THz device capable of making measurements was constructed for use outside. Sub-terahertz band devices are compact and harmless to humans, so the constructed device could be carried from the laboratory to a public environment for measurement. The smart identification equipment shown in Figure 9a is a PP and PE material identification system for plastic bottle caps. These devices were placed in a 700 mm × 500 mm × 1200 mm box. The measurement frequency for the PET bottle caps was 0.14 THz.

The waste caps were placed on the stage through a special cap window; they are designed to fall between the terahertz transmitter and receiver. A cap’s centre of gravity is on its surface, so the top of the cap was placed in full contact with the stage, and thus, the interior was face-up. This arrangement made it possible to irradiate the cap with sub-THz waves at the appropriate measurement position. The cap was then irradiated with terahertz waves from above, and the transmission intensity information was transmitted to a PC, which selected the material based on approximate values from a database and sent a signal to the PP- or PE-sorting mechanism. The transmittance database was already available on the PC. Afterwards, the sorting mechanism, via an air jet sorting system, fed each material into a collection box.

As for other mechanisms, the THz device’s identifier details include a box for PET bottles (material: PET) and bottle labels (material: PS); notably, hand sorting is still required. In addition, to make it easier to put waste bottles in the boxes, questions were displayed at the entrance of the plastic bottle disposal area, and the disposer would respond by placing the bottle in either the ‘left (yes)’ or ‘right (no)’ boxes. This equipment identifies and sorts the PET bottle caps using sub-THz wavelengths and collects the PET bottles according to the material type. This system was installed at the entrance and near the canteen to ensure its use.

The results confirmed that the caps of plastic bottles for beverages could be identified as being composed of either PE or PP with a discrimination rate of more than 90%. The difference in the experimental results was attributed to the dirtiness or inner ribbing of the samples, resulting in a low identification rate. In the future, we aim to incorporate such sample conditions into machine learning to improve the accuracy rate.

## 8. Discussion

Based on spectral information acquired via a GaP terahertz spectroscopy measurement system, which can sweep frequencies from 0.5 THz to 7 THz, and a sub-terahertz band measurement system, which uses small, inexpensive semiconductor electronic devices at specific frequencies below 0.14 THz, a smart device capable of differentiating between types of waste plastic was constructed and tested. The sub-terahertz band devices employed are compact and harmless to humans, so the constructed device could be carried from the laboratory to a public environment to acquire measurements. 

With reference to the spectroscopic data in the terahertz wave band that were conventionally measured and databased, a practical measurement device for the sub-THz band, especially suited to the identification of PET/PS and PP/PE, was developed. This system is designed to be unaffected by anisotropy due to molecular chain orientation. The performance and validity of the system, as well as its instrumentation, were verified. A spectrometer operating in the sub-terahertz band proved to be effective for the identification of PP/PE and PET/PS based on molecular weights and could contribute to improving the efficiency of the material-recycling process. As well as improving the efficiency of recycling, it also reduces the environmental impact of the waste management process. Specifically, quick and accurate sorting of each type of plastic improves the quality of the recycling process, ultimately leading to higher recycling rates and less waste.

Smart cities need to manage all public services effectively, including waste management. However, it is not easy to collect data and analyse recycling processes because existing waste management and recycling systems do not consider the use of private sectors and IT. In this paper, we proposed a new concept with the potential to create a smart waste management and recycling system. This practical measurement device, being both small and portable, operating in the sub-THz band can collect data. If the collected data are aggregated in the cloud, this device could become a very powerful tool for pre-sorting before waste collection is available in supermarkets close to consumers and revolutionise recycling technologies in smart cities. In addition, this technology is not limited to urban areas; it can be applied to a wide range of areas, including waste treatment plants.

Furthermore, data collected through operations in supermarkets and treatment plants could be aggregated in the cloud to further optimise the recycling process, and aggregating these data in the cloud has the potential to further optimise the recycling process and enable a revolution in recycling technology in smart cities. We also believe that the data can be managed and acted upon, as new plastics and composites are discarded every day. In addition, it is hoped that the increasing popularity of this technology will lead to a more sustainable and efficient society.

## 9. Conclusions

In this study, a smart material identification system was developed to identify plastic waste (PET, PS, PP, and PE) at the point of disposal. This system reduces the sorting workload for recyclers. Specifically, terahertz (0.5–7 THz) and sub-terahertz (<0.14 THz) spectroscopy were shown to be effective in identifying the type and molecular weight of plastics. This capacity is essential for proper recycling. In addition, a smart waste management system using portable sub-THz devices for pre-sorting in consumer environments has been proposed to improve pre-collection recycling rates in smart cities. Material identification at the point of disposal supports sustainable recycling and decarbonisation efforts relating to polymers, reducing contamination by other materials and improving recycling quality. Additionally, advanced technologies such as THz spectroscopes can facilitate the development of more automated and innovative waste management systems. Our system achieved almost 100% accuracy in identifying PP and PE lids and 93% accuracy in identifying PET and PS containers and packaging plastics.

The sub-THz-band-based waste plastic identification technology developed in this study has the potential to give rise to innovative improvements in waste management and recycling processes. This sub-THz spectrometry system is portable, enables plastic identification, and streamlines the process of sorting individual plastics. These features are expected to accelerate waste disposal and increase recycling rates, contributing to a reduction in environmental impacts. In addition, this technology could be an important step towards reducing landfill waste and creating smart cities. In particular, if used as a pre-sorting system in supermarkets and shopping centres, it could maximise the effectiveness of recycling by efficiently sorting waste before it is collected. In addition, cloud-based data aggregation could enable real-time optimisation of waste management and recycling, improving the overall efficiency of waste management.

In the future, this technology will be further developed and applied to other sectors, making it an important driver of innovation in waste management. This is expected to contribute to the realisation of a sustainable and efficient society, significantly contributing to environmental protection and the realisation of a resource-recycling society. In addition, improvements will be made to overcome the limitations of current technology, such as miniaturisation, and to be able to manage more types of plastics and waste. In addition, the system will be extended to other applications, with the aim of contributing to the efficiency and environmental protection of society as a whole.

## Figures and Tables

**Figure 1 polymers-17-00462-f001:**
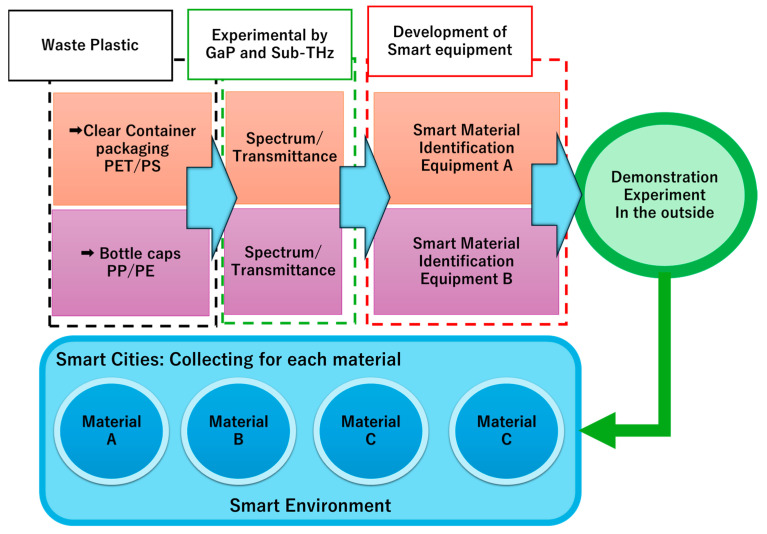
Research background and methodology.

**Figure 2 polymers-17-00462-f002:**
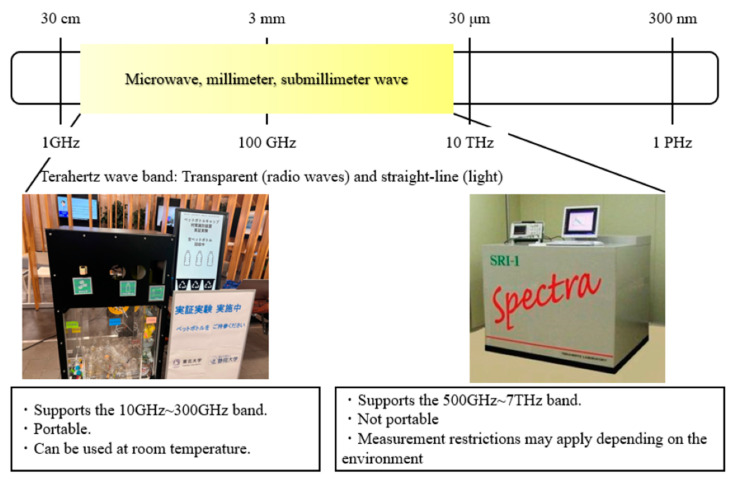
Broadband terahertz spectroscopic equipment (the right picture was adopted from [30]).

**Figure 3 polymers-17-00462-f003:**
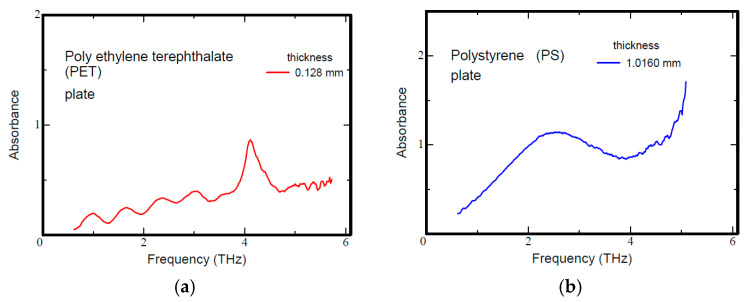
Terahertz spectra of polyethylene terephthalate (**a**) and polystyrene (**b**).

**Figure 4 polymers-17-00462-f004:**
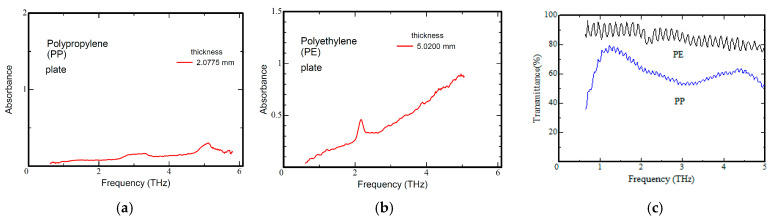
Terahertz spectra from 0.5 THz to 5 THz of (**a**) a polypropylene plate, (**b**) a polyethylene plate, and (**c**) PE and PP bottle caps [15].

**Figure 5 polymers-17-00462-f005:**
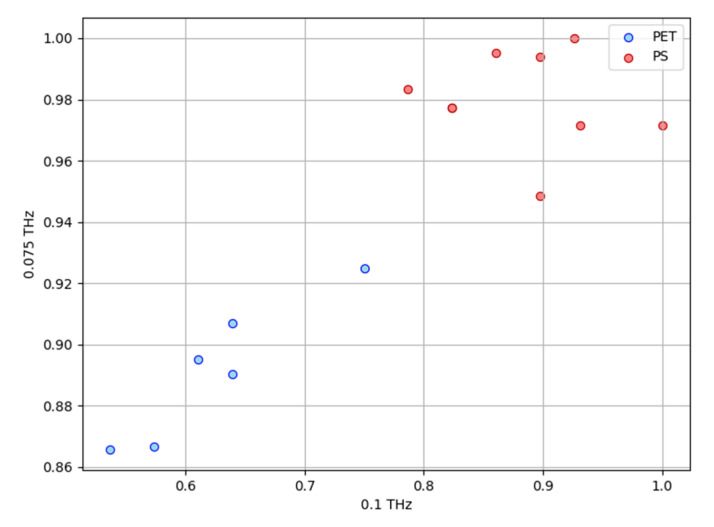
Terahertz transmission distribution of PET and PS at frequencies of 0.1 THz and 0.075 THz.

**Figure 6 polymers-17-00462-f006:**
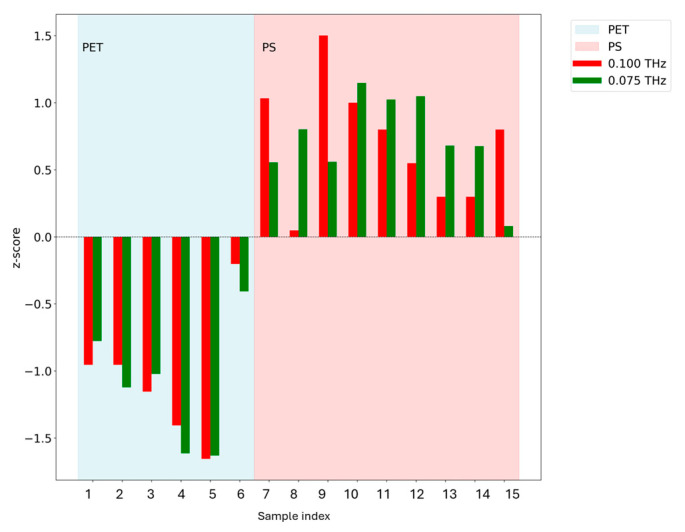
The z-scores for the transmittance rates of each sample.

**Figure 7 polymers-17-00462-f007:**
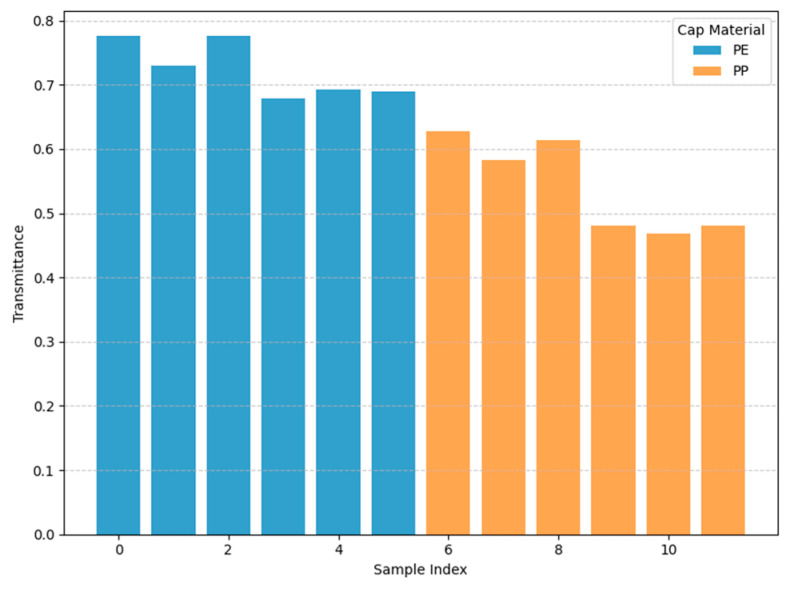
Bottle caps’ transmittance rates after being subjected to 0.14THz irradiation.

**Figure 8 polymers-17-00462-f008:**
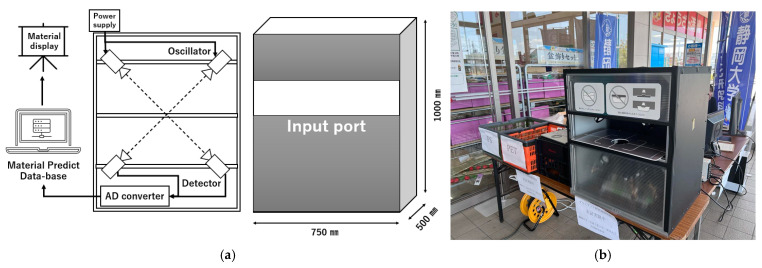
The design of the smart identification equipment for identifying waste plastic containers used in the demonstration (**a**), and the implementation of the demonstration (**b**).

**Figure 9 polymers-17-00462-f009:**
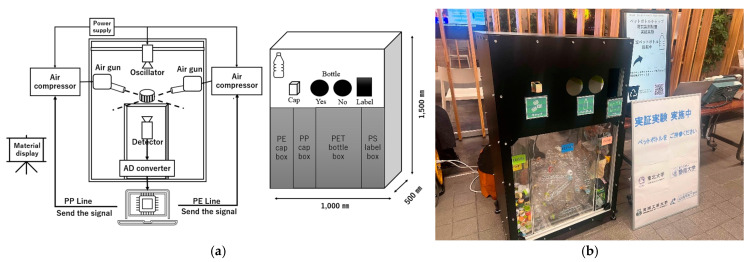
The design of the smart identification equipment for identifying waste bottle caps used in the demonstration (**a**), and the implementation of the demonstration (**b**).

**Table 1 polymers-17-00462-t001:** Transmittance of PP and PE bottle caps when applying a frequency of 0.14 THz.

Cap Material	Hight (mm)	Colour	BG (mV)	Transmitted Value (mV)	Transmittance
PE	7.5	White	343	266	77.6%
PE	7.5	White	343	250	72.9%
PE	7.5	White	343	266	77.6%
PE	13.5	White	343	233	67.9%
PE	13.5	White	342	237	69.3%
PE	13.5	White	342	236	69.0%
PP	8	White	343	215	62.7%
PP	8	White	343	200	58.3%
PP	8	White	342	210	61.4%
PP	14.5	White	342	164	48.0%
PP	14.5	White	341	160	46.9%
PP	14.5	White	341	164	48.1%

## Data Availability

All data are presented in this manuscript.

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
