# Peer review of "Development of Smart Material Identification Equipment for Sustainable Recycling in Future Smart Cities"

_polymers, 2025, doi:10.3390/polym17040462_

Round 1
Reviewer 1 Report
Comments and Suggestions for Authors
The manuscript entitled “Development of Smart Material Identification Equipment for Sustainable Recycling in Future Smart Cities” is interesting and has the potential to be considered for publication. However there are few major changes that need to be carried out before acceptance. The major drawback is the discussion session that needs to be more elaborated. Other comments are as follows.
-
Highlights are two descriptives that make it as small points with limited characters. Avoid sub headings under highlights.
-
Add quantitative data in th abstract
-
Keywords: Include some sustainability development related keywords
-
Expand the abbreviations in the first place of appearance within the text.
-
Novelty factor of the work is not clearly mentioned in the introduction. Properly justify how this work is different from other works.
-
Fig 4: Specify at which wavelength the absorbance is measured.
-
What is the type of statistical analysis done to validate the data?
-
Discussions are too short. Author and team should share their results with the existing literature available . In its present form discussion is lacking clarity.
-
Present conclusion as a separate section and include the future prospects of the work.
-
Reference [4,5,6] to be converted as [4-6]. Check this throughout the manuscript
-
Reference style is not uniform, correct it as per the journal guidelines
Author Response
Thank you for reviewing our work. Please refer to the attached file for our responses.

Reviewer 2 Report
Comments and Suggestions for Authors
Attached you can find my comment.

Author Response
Thank you for reviewing. Please refer to the attached file for our responses.

Reviewer 3 Report
Comments and Suggestions for Authors
The development of the smart material identification equipment in this study, aimed at achieving sustainable recycling in future smart cities, has primarily focused on the creation of a smart material identification system. This research has led to the development of a smart identification system for typical plastic wastes such as PET, PS, PP, and PE, utilizing sub-terahertz (sub-THz) technology to identify materials at the point of waste disposal, thereby significantly reducing the burden on recyclers. The equipment effectively identifies different plastic materials through terahertz (THz) and sub-terahertz spectroscopy, recognizing the molecular weights and types of typical plastic wastes. The study proposes a new concept for a smart waste management and recycling system, which leverages data from portable sub-THz devices to better pre-sort materials in consumer-facing environments, such as supermarkets, before waste collection. This has the potential to revolutionize recycling technologies in smart cities.
The research demonstrates that the use of advanced material identification technologies and cloud-based data aggregation can enhance the quality and efficiency of recycling typical plastic wastes in smart cities, paving the way for more automated and technologically advanced material management. This study significantly promotes the application of advanced material identification technologies and cloud-based data aggregation to improve recycling efficiency in smart cities, with strong practical significance and application prospects, as well as high practicability and value. However, there are still some issues in the manuscript that need to be revised and perfected, and it is recommended to publish after revision. The issues that need to be revised are as follows:
1.The literature format in the text needs to be carefully checked, and periods should be added to the ends of sentences for references 2-10, 14, 15, 17-21, 26, and 30.
2.Although the equipment performs well in laboratory and supermarket environments, it is recommended to further improve its environmental adaptability by testing its performance under different environmental conditions, such as changes in humidity and temperature.
3.It is suggested to supplement technical details in the paper, including more detailed descriptions of the equipment's specific technical parameters and operating procedures, which would be beneficial for reference and reproducibility.
4.Ensuring the chemical stability of samples is an important aspect of experimental design, especially in studies involving material identification and spectroscopic analysis, as it directly affects the accuracy and reproducibility of experimental results. Therefore, the text should include conditions for plastic test samples, such as sample purity and pre-treatment conditions.
5.It is recommended to add experimental data comparing the accuracy of identifying different batches to ensure the stability of the materials. Including this discussion would enhance the paper significantly.
6. It is suggested to improve the diversity of experimental samples and enhance the robustness of the system by expanding the range of waste plastic materials that the smart material identification equipment can recognize. In addition to PET, PS, PP, and PE, it is recommended to include other common waste plastic materials with high recycling value, such as PVC, ABS, HDPE, polycarbonate (PC), and polyurethane (PU).
Comments on the Quality of English LanguageThe English could be improved to more clearly express the research.
Author Response
Thank you very much for your thorough review. Please find our responses in the attached file.

Reviewer 4 Report
Comments and Suggestions for Authors
Manago and co-workers have developed an approach to identify different polymeric materials with the purpose to streamline the recycling process, particularly the sorting step. The idea is interesting and demonstrated the authoring thinking on applying science to real-world applications. The manuscript overall is well constructed and logical in communicating the findings from the experiments.
There are few suggestions or questions that I would like to highlight to the authors:
1. In reality, the plastic packaging is not just purely polymeric materials, there are different additives, dye, or in certain case blending of different molecular weights, would any of those "impurities" impact the result? Also, would the shape of the material, e.g., thickness, impact the effectiveness of the equipment?
2. Many of the plastic wastes are by definition composite, i.e., not all the scenarios would be as simple as plastic bottles, how would such technique perform when trying to sort those materials, e.g., a waste comprising both parts in PET and PE?
3. The authors indicated that the technology is for smart city use, I understand this could be related to smart city, but the nature of such technology is not really limited to recycling in smart cities, it could in industry scenario as well, so I would suggest the authors to broaden the scope of application if applicable
Comments on the Quality of English LanguageThe manuscript is relatively well written, there are opportunities to de-word a bit.
Particularly, the authors used about half of the manuscript to explain the context of the research and discuss prior work, which is excessive and not necessary, I would suggest the authors to streamline the introductory sections and get to results earlier.
Author Response

(The authors gave the same response as above.)
